# Imaging Features of Plantar Vein Thrombosis: An Easily Overlooked Condition in the Differential Diagnosis of Foot Pain

**DOI:** 10.3390/diagnostics14020126

**Published:** 2024-01-05

**Authors:** Frederico Celestino Miranda, Adham do Amaral e Castro, Fábio Brandão Yoshimura, Alexandre Leme Godoy-Santos, Durval do Carmo Barros Santos, Laercio Alberto Rosemberg, Atul Kumar Taneja

**Affiliations:** 1Hospital Israelita Albert Einstein, São Paulo 05652-900, Brazil; fcmmsk@gmail.com (F.C.M.); adham.castro@gmail.com (A.d.A.e.C.); fabo.yoshimura@gmail.com (F.B.Y.); alexandrelemegodoy@gmail.com (A.L.G.-S.); durvalcs@einstein.br (D.d.C.B.S.); laercio.rosemberg@einstein.br (L.A.R.); 2Department of Imaging Diagnosis, Universidade Federal de São Paulo, São Paulo 04024-002, Brazil; 3Faculdade de Medicina, USP, São Paulo 05403-010, Brazil; 4Department of Radiology, UT Southwestern Medical Center, Dallas, TX 75390, USA

**Keywords:** thrombophlebitis, deep venous thrombosis, foot pain, MRI, Doppler, DVT, metatarsalgia

## Abstract

Plantar vein thrombosis is a venous disorder affecting deep plantar veins that can manifest with non-specific localized pain, plantar foot pain, swelling, and sensation of fullness. Plantar veins are not routinely assessed during sonographic scans for deep venous thrombosis, which makes plantar venous thrombosis a commonly missed diagnosis. This paper provides a comprehensive review of the venous anatomy of the foot and imaging findings of plantar venous thrombosis as well as discusses the current literature on the topic and its differential diagnoses.

## 1. Introduction

Deep vein thrombosis of the lower limbs is an extensively researched topic with well-established clinical guidelines [1]. However, when it comes to plantar vein thrombosis (PVT), there is no defined consensus on diagnosis or treatment, and it is an underdiagnosed condition with few cases reported in the literature [2,3]. The clinical diagnosis is usually challenging, with the main differentials including conditions that are more common to cause plantar pain and metatarsalgia, such as plantar fasciitis, tendinopathy, ganglion cysts, crystal deposition disease, Morton’s neuroma, intermetatarsal bursitis, sesamoiditis, plantar plate injuries, and stress fractures [2,4].

Ultrasonography (US) is the method of choice for diagnosing thrombosis in general, but plantar veins are not routinely encompassed in the lower limb venous scanning protocols, and ultimately magnetic resonance imaging (MRI) has shown to be a reliable resource in clinical practice [1,2,4]. Although imaging findings are well known for deep venous thrombosis (DVT) and include venous filling defects, venous ectasia, and perivascular edema and enhancement, dedicated studies for PVT are scarce case reports and case series [4].

The exact prevalence and incidence of PVT is unknown [5], but some studies suggest it accounts for about 10% of the patients with DVT, compared to about 44% in the calf veins [6]. The lateral plantar vein is affected in most of the cases (96%), while the medial plantar vein is less frequently affected (27%) [7]. In our experience, concurrent involvement of the plantar arch and plantar metatarsal veins can be frequent.

The present article will address PVT in a didactic and illustrative manner, reviewing plantar venous anatomy, main imaging findings, and its differential diagnoses.

## 2. Anatomy

Foot venous anatomy is complex and highly variable, with many variations described in the literature. The following discussion includes the most important plantar veins visualized on routine MRI scans with a distal to proximal approach.

The plantar digital veins originate from the plexus on the toes, joining to form the metatarsal veins, located in the metatarsal spaces, which then form the deep plantar venous arch, located at the level of the proximal forefoot. They drain into the medial and lateral plantar veins, accompanying the lateral and medial plantar arteries, which, after emitting the great and small saphenous vein, unite behind the medial malleolus to form the posterior tibial veins [4,8,9]. Of note, the deep plantar venous arch, metatarsal veins, and medial and lateral plantar veins lie deep to the plantar muscle groups, below the osseous structures of the foot, with the deep plantar arch and metatarsal veins situated deep to the oblique and transverse heads of the adductor hallucis muscle. The lateral plantar vein is positioned between the flexor digitorum brevis muscle and quadratus plantae. The medial plantar vein courses between the abductor hallucis and the flexor hallucis brevis muscles [9,10]. The deep plantar venous anatomy is shown in Figure 1.

The literature suggests that lateral plantar veins are more frequently affected by thrombosis than the medial plantar veins [4,7], which is also in line with the authors’ experience. This could be due to the proximity of the lateral plantar veins to the sole of the foot, which can make them more susceptible to mechanical stress. On the other hand, medial plantar veins are smaller, and thrombosis could be more difficult to be appreciated for this reason.

Also important are the perforating veins of the foot that make connections between the deep and superficial veins at the superficial dorsal venous arch, acting as an ascending venous pump [11,12,13]. In the foot, perforating veins show distinctive features from lower limb veins, with some of them being valveless and allowing bidirectional flow or presenting inverted valves and enabling flow from deep to superficial veins [11]. For this reason, and from the hemodynamic perspective, foot veins should not be classified as deep and superficial systems, but rather as medial and lateral anatomical/functional units [11]. Medial perforators of the foot directly connect the deep veins (medial plantar veins) to the superficial veins (medial marginal veins), forming a unique “medial functional unit” that directs the venous flow from deep to superficial [11]. According to Rastel et al., the first interspace metatarsal perforator was the perforator most frequently affected in PVT [5]. This perforator is the major connector between the deep veins and the superficial veins and could be the origin of the PVT [11].

The plantar venous system is an important vascular pump for the leg [14]. The blood reservoir of the foot is deeply located in the plantar veins, between the plantar muscles. The medial and especially the lateral plantar veins converge in the calcaneal region, where the blood is ejected upward into the posterior tibial veins. During walking, with each step, an estimate of 25 mL of blood is mobilized upward [11].

## 3. Predisposing Factors

The pathogenesis for PVT is still uncertain and can be idiopathic or related to multiple causative systemic or local factors [8,15,16]. The main underlying factor for the development of deep venous thrombosis is the well-known “Virchow triad” formed by venous stasis, endothelial damage, and inflammation [16], with possible predisposing factors including recent surgery, trauma, infection, malignancy, airplane travel, oral contraceptives, mechanical stress, athletic activity, post-operative immobilization, coagulation disorders, and pressure from orthotics [2,8,15].

Increased mechanical load or stress to the plantar region of the foot appears to be a very typical presentation of PVT [17], differently from other frequent DVT in other locations. Mechanical strain, including the use of footwear and orthoses [17,18], could be a unique risk factor, due to repeated microtrauma of the veins, with activation of the coagulation cascade. In a series of 22 patients published by Czihal et al., the majority of cases were idiopathic, but history of mechanical strain to the foot was present in one-third of the patients [7]. This could also explain why PVT is more frequent in the lateral plantar veins, which have a closer proximity to the sole of the foot when compared to the medial plantar veins. In addition, there is a high level of recurrence, reaching up to 27% according to Czihal et al. [7]. Recently, infection by the Coronavirus disease of 2019 (COVID-19) has also been implied as a possible predisposing factor but has yet to be deeply studied [19,20].

## 4. Clinical and Laboratorial Findings

PVT usually manifests as non-specific “heaviness or fullness” sensation or localized plantar pain, similar to neuroma or plantar fasciitis, the latter being a great mimicker [2]. Swelling and pain that increases during walking are also noted symptoms at presentation [5,8]. The diagnosis of PVT as a cause of foot pain is rarely suspected by the clinician or orthopedist in the clinical setting [21]. Time from symptoms onset to diagnosis can take up to four weeks, being one week in most of the cases [5]. This can occur because of initial misdiagnosis but also due to the low intensity of pain, delaying medical care and explaining a more extensive thrombus in some cases. Fortunately, symptomatic pulmonary embolism is a rare complication, differently from DVT [15].

Laboratory tests include D-dimer levels, which are known to be highly sensitive but not very specific, with a high negative predictive value (NPV) for the presence of DVT [16]. Measuring D-dimer in the acute setting in patients at low risk of DVT (assessed clinically) can help exclude this diagnosis. In regards to other laboratory tests in the acute setting, studies show that lupus anticoagulant and antiphospholipid antibodies may also be useful for the diagnosis of DVT [22].

## 5. Imaging Evaluation

The most common imaging method used to evaluate plantar venous thrombosis is ultrasound (US), and this may be applied to plantar vein thrombosis as well. Regarding PVT, in general, the scanning protocols for thrombosis of lower limbs are limited to the distal leg and do not extend to the foot [1,2]. Thomas and O’Dwyer recommend the inclusion of at least one view of the foot veins in a standard phlebogram in patients with suspicion for DVT or pulmonary embolism [6]. Findings with US are similar to that of a DVT in other segments, including loss of venous compressibility, venous dilatation, lack of flow on Doppler study, and intraluminal content leading to filling defects (Figure 2, Figure 3 and Figure 4) [1,2,4,8].

Forefoot PVT is difficult to evaluate using ultrasound due to the proximity of osseous structures and thickness of plantar skin and subcutaneous layer, causing artifacts and attenuation of the US beam, especially in patients with obesity and thickened skin [8,13]. Even with targeted Doppler ultrasound examination of the forefoot veins, sensitivity can be low due to small size of the vessels and anatomic variations [23]. For this reason, in our experience, MRI has shown to be particularly helpful in evaluating thrombosis of the foot, thus radiologists should actively assess plantar veins on every forefoot and ankle MR reading.

Computed tomography (CT) in the context of venous thrombosis is commonly used to evaluate for pulmonary embolism or larger vessel involvement such as inferior vena cava and iliac veins [24]. Although it can be considered as the gold standard for venous thrombosis in general, venography studies in the context of plantar venous thrombosis are lacking, since US is the method of choice, with good sensitivity and specificity compared to venography [1]. Therefore, MRI studies come as a major diagnostic tool for this condition [2], due to its higher detail and resolution, with imaging findings including perivascular edema, muscular edema, intravascular heterogeneous signal intensity, venous ectasia, presence of collateral veins, perivascular enhancement, and venous filling defects [4].

The most prominent finding is perivenous edema and enhancement, corresponding to inflammatory soft tissue changes [21]. These inflammatory changes are caused by a combination of venous congestion and inflammatory response to the thrombus, with edema and enhancement extending through various muscle compartments along the involved vein course [21,25]. Close inspection of the venular bundles is essential in cases of an unexplained muscle edema to confidently exclude a possible diagnosis of PVT.

Intraluminal signal intensity is variable and can be decreased or increased on T1 or T2-weighted images due to different imaging characteristics of blood flow and its products. The intermediate or decreased luminal signal intensity can cause a “vanishing vessel sign”, making the vein difficult to differentiate from the adjacent muscles. Intravascular signal is also dependent on inflow and washout effects caused by the motion of high-intensity unsaturated blood in the plane of acquisition and loss of signal due to outflow of excited protons from the imaging plane before applying refocusing pulse [25]. That is the reason why faster flowing blood in the center of the vein can result in a hypointense signal compared to the high signal intensity of the peripheral slower flow on the axial slice (“target appearance”). This should not be considered a thrombus. When a thrombus occludes a vein, the normal hyperintense T2-weighted image signal of a slow-flowing blood is replaced by a hypointense thrombus [25].

Intravenous contrast is very helpful in detecting PVT, especially in the forefoot, where the veins are smaller, increasing the conspicuity of the findings, outlining the intramuscular structures through the enhancement of vascular walls with vasa vasorum [23]. After contrast administration, decreased signal intensity can be observed in the affected vessel lumen delimitating the thrombus as a filling defect and confidently diagnosing the thrombosis. If the thrombus is not visualized after the injection of contrast, the possibility of only inflammatory changes (phlebitis) without thrombosis should be considered. In such cases, targeted ultrasound to the suspected area can aid to confirm the presence of a thrombus vs. “flow void” effect caused by the fast blood flow. Given the small size of the clots, in some cases, it is unlikely that even focused duplex ultrasound assessment could show the thrombus when compared to contrast-enhanced MRI study [23]. Table 1 shows the main MRI and US findings (B-mode and Doppler) for the diagnosis of PVT.

Thrombosed plantar veins can present with either normal caliber or enlarged. Dilatation of the plantar veins can be a common finding in patients without venous thrombosis, as a variant of normality or secondary to venous insufficiency and varicosities. Therefore, in our experience, this finding should not be used as a reliable diagnostic criterion.

As per the authors’ experience, the increased use of MRI to assess foot pain in the clinical workflow has led to PVT being frequently diagnosed first on MRI, with the method also being helpful to exclude other causes of plantar pain [13]. One of the critical points to be emphasized, based on the literature reviewed and the authors’ experience, is the use of US in the context of diagnosing PVT, where this diagnosis is missed in two common scenarios. The first scenario is the search for deep vein thrombosis of the leg using Doppler US protocol. If there is pain in the foot, although it is not part of the routine protocol, it is important to extend the evaluation to the veins of the foot. Another scenario is the routine US scan of the ankle and/or forefoot in the assessment of metatarsalgia. It is important to increase the awareness of this important and not infrequent diagnosis to allow the examiner to actively assess foot veins. Of note, although US is an excellent method for most cases, the examiner’s experience and knowledge of this diagnosis must be considered. While assessing metatarsalgia using MRI, this diagnosis is generally not missed by skilled radiologists, even though this method is less available and more expensive than US worldwide.

## 6. Differential Diagnosis

There are numerous differential diagnoses for forefoot pain, as clinical findings are often nonspecific, with complaints sometimes being broad and vague, or even in locations where other conditions are more common. These include plantar fasciitis, plantar fibromatosis, tendon pathologies, ganglion/synovial cysts, crystal deposition disease, Baxter neuropathy, Morton’s neuroma, intermetatarsal bursitis, sesamoiditis, plantar plate injuries, and metatarsal stress fractures [4,26]. Table 2 shows the main differential diagnoses with clinical presentation and the main imaging findings on MRI and US. In fact, commonly, the possibility of PVT diagnosis is not raised due to lack of awareness and familiarity with the entity, and this is reflected clinically and in the literature [27].

Metatarsalgia diagnostic reasoning, strategy, and subsequent management involve a series of clinical steps to arrive at the cause and the appropriate treatment [28]. Due to its lower frequency, PVT is often not included among the differentials for foot pain. And considering that PVT can occur in basically any region of the foot, depending on where it is located, it has the potential to simulate various causes of foot pain. Thus, differential diagnoses are related to surrounding anatomical structures. For example, thrombosis of the lateral plantar vein may simulate plantar fasciitis and peroneal tendinopathy. Medial plantar vein thrombosis, on the other hand, may mimic posterior tibial tendinopathy. Thrombosis affecting the plantar venous arch, metatarsal veins and digital veins can have similarities with metatarsal stress fracture, Morton’s neuroma, intermetatarsal bursitis and sesamoiditis. 

### 6.1. Intermetatarsal Bursitis

Intermetatarsal bursitis is a common cause of metatarsalgia and frequently associated with Morton’s neuroma and plantar plate tears. It is characterized by fluid distension of the bursa between the metatarsal heads. Bursal distension smaller than 3 mm in the transverse diameter may be physiological and not clinically relevant [29].

Ultrasound findings in acute bursitis include a thin-walled bursa distended with hypoechoic fluid and peribursal hyperemia. In chronic bursitis, a thickened bursal wall, synovial proliferation, more echogenic content, and intra-bursal hyperemia can be present. Sometimes, chronic intermetatarsal bursitis can simulate soft tissue mass or abscess if marked synovial thickening is seen. On MRI, intermetatarsal bursitis is hypointense on T1-weighted and hyperintense on T2-weighted fat-suppressed sequences. Peripheral enhancement may be seen with gadolinium administration [27]. Figure 5 exemplifies a case of intermetatarsal bursitis.

### 6.2. Morton’s Neuroma

Morton’s neuromas are focal areas of symptomatic perineural fibrosis around the plantar digital nerve of the foot. It does not represent a true neuroma [29], and the most accepted hypothesis is thought to be related to chronic entrapment of the nerve by the intermetatarsal ligament [30], leading to perineural fibrosis [31]. They occur more often in middle-aged women [31]. The third intermetatarsal space is the most affected site, followed by the second intermetatarsal space, and the remaining spaces are rarely involved. Larger lesions (>5 mm) tend to be more symptomatic [29].

Ultrasound findings include well-defined ovoid mass with variable echogenicity and continuity with the interdigital nerve in the long axis. The mass can be tender and mobile when compressed, with vascularity on power Doppler [27]. The sonographic Mulder sign can be used to increase diagnostic accuracy, with compression of the metatarsal heads leading to exposure of the neuroma and a possible click felt by the examiner [32].

MRI findings include an ovoid or teardrop-shaped mass in the plantar aspect of the intermetatarsal space, located inferiorly to the intermetatarsal ligament, that is most clearly visualized in the prone position [31]. The mass is generally hypointense to isointense on T1-weighted images and hypointense to hyperintense on T2-weighted fat-suppressed images. Signal intensity can vary according to the degree and maturity of the fibrosis. Post-contrast images can improve the visualization of the neuroma, which can enhance variably, with most demonstrating little to no enhancement [27]. When enhancement is present, it is generally due to the enhancement of the bursal tissue surrounding the neuroma [27]. The decision to inject the contrast medium for this diagnosis is controversial [33]. Figure 6 illustrates a case of Morton’s neuroma.

### 6.3. Sesamoiditis

Sesamoiditis is a painful mechanical-related inflammatory condition involving the sesamoid bone caused by repetitive injury to the plantar aspect of the forefoot, with this term being used almost exclusively for the hallux sesamoid. It may manifest as an acute entity presenting with bone marrow edema of the sesamoid, while in chronic phase, it will show volumetric decrease and sclerosis. Medial sesamoids are more commonly injured than the lateral [34]. Other conditions affecting the hallux sesamoids are osteonecrosis and trauma, the latter leading to sesamoiditis, acute fracture, or diastasis of a bipartite sesamoid [34]. Figure 7 illustrates a case of sesamoiditis of the forefoot.

### 6.4. Plantar Fasciitis

The plantar fascia is a thick band of connective tissue that originates in the medial tubercle of the calcaneus and inserts in three different places in the forefoot, thus creating three distinct bands: medial, central, and lateral [35].

Plantar fasciitis (PF) is the most common cause of unilateral heel pain [2] and refers to the inflammation of the fascia of the foot. On ultrasound, PF appears as thickening and hypoechogenicity of its fibers. MRI is considered the most sensitive imaging modality for diagnosing PF [36], characterizing the exact location and the extent of alterations.

Ultrasound is reliable and accurate to assess the plantar fascia, and the longitudinal scan is the best plane for imaging [37]. Ultrasound findings include thickening greater than 4.5 mm (the most useful sign), hypoechogenicity of the plantar fascia, and loss of normal fibrillar reflective echotexture. Increased stiffness on elastography has been reported [38]. A plantar calcaneal spur is commonly seen deep to the proximal fascia [37].

MRI findings of plantar fasciitis include hypointense or isointense fascial thickening at its calcaneal insertion on T1-weighted images, with its thickness greater than 4.5 mm and in most of the cases associated with subcutaneous or perifascial edema [38]. Insertional bone marrow edema can be present in some of the cases and may be related to mechanical changes or inflammatory enthesitis [39]. Figure 8 presents main findings of plantar fasciitis.

### 6.5. Plantar Fibromatosis

Also known as Ledderhose disease, plantar fibromatosis is a rare and benign entity presenting with ill-defined infiltrative heterogeneous masses in the deep aponeurosis (plantar fascia), adjacent to the plantar muscles. These nodules can be locally aggressive. This condition has an increased prevalence in men and is more frequent in patients with diabetes and epilepsy. Ultrasound has advantages over MRI in the assessment of plantar fibromatosis, since fibromas are easy to detect on US due to the contrast between the poorly reflective fibroma and the fibrillar appearance of the normal fascia, as well as the possibility of direct contralateral comparison [30,36].

Ultrasound findings include diffuse fusiform-shaped nodules of plantar fascia separated from the calcaneal insertion that are hypoechoic or isoechoic to the plantar fascia. Either posterior acoustic enhancement or shadowing can be seen, as well as hypervascularity on Doppler. The majority are located within midsubstance or superficial to the plantar fascia. If large with infiltrative margins, consider aggressive plantar fibromatosis [30].

MRI findings include isointense plantar fascia nodules on T1-weighted images and on T2-weighted fat-suppressed images, with contrast enhancement after the administration of gadolinium [30]. A case of plantar fibromatosis is demonstrated in Figure 9.

### 6.6. Tendon Pathologies

Foot tendon pathologies are common and seen in a wide range of patients from young athletes to older patients. These pathologies include tendinosis (tendinopathy), tenosynovitis and peritendinitis, partial and complete tears, subluxation and dislocations, and, rarely, tendon entrapments. Tendinosis represents tendon degeneration and manifests as enlargement and increased intrasubstance signal on T2-weighted images. Tendon tears appear with higher signal intensity or even fluid signal intensity on T2-weighted images. Partial tears may show tendon enlargement due to longitudinal splits or thinning, caused by partial disruption of the tendon fibers. Full-thickness tears are represented by complete disruption, generally with retraction and tendon gap. Tenosynovitis is the inflammation of the tendon sheath, with an increased amount of fluid and inflammation of the surrounding soft tissues. Peritendinitis is the inflammation of the peritenon, and paratendinitis refers to inflammation of the adjacent tissues [40,41,42].

Ultrasound demonstrates intra and peritendinous alterations, with increase in tendon thickness and different degrees of hypoechogenicity. MRI demonstrates enlargement and signal alterations of the tendon affected. High signal intensity can be seen involving the tendon and peritendinous soft tissues. Insertional tendinopathy (enthesopathy) can be associated with bone spurs and calcifications [40,41,42]. Tendon pathologies are presented in Figure 10 with illustrative images.

### 6.7. Ganglion/Synovial Cysts

Synovial cysts are caused by the herniation of the synovial membrane through the joint capsule, and typically there is a persistent communication with the joint with synovial tissue lining. Ganglion cysts are a discontinuous layer of pseudo-synovial cells, surrounded by non-synovial connective tissue and not always communicating with the joint. They may be in or associated with joint capsules, ligaments, tendon sheaths, bursa, or bone [29].

These cysts are easily seen on ultrasound as well defined, uni, or multilocular fluid-filled anechoic masses, with posterior acoustic enhancement. The stalk can be identified, indicating a communication with the adjacent joint or tendon sheath of origin [27].

On MRI, the cyst can be seen as a mass with water equivalent signal, with uniformly hyperintense and the walls can show enhancement after gadolinium administration. Narrow stalk connecting cyst to joint can also be visualized [29]. Figure 11 shows a case of ganglion/synovial cyst.

### 6.8. Stress Fractures

Stress fractures occur due to a mismatch of strength and mechanical stress on the bone. They are the most common bony cause of metatarsalgia [33]. Fatigue stress fractures occur in athletes, especially runners and military recruits, by overload of normal bone. Insufficiency stress fractures are more common in women and older patients, even with normal loads on demineralized bone [27]. Metatarsal shafts are the most common site for stress fractures, especially of the second or third rays [43]. Initial radiographs can be negative for up to 2 to 3 weeks.

Although ultrasound has low sensitivity to detect stress fractures, it can show thickening and hypervascularity of the periosteum, cortical irregularities, and subcutaneous edema [34]. Cortical fracture lines and callus formation are better evaluated by radiographs [27]. MRI is the modality of choice, demonstrating periosteal thickening and bone marrow and surrounding reactive soft tissue edema [43]. MRI findings include jagged and irregular, incomplete or complete low signal fracture line through bone, characterized by a band of low signal intensity contiguous with the cortex on both T1 and T2-weighted images. The bone marrow edema may be feathery, stellate, or band-shaped [27]. Figure 12 nicely demonstrates an example of stress fracture.

### 6.9. Plantar Plate Injuries

Plantar plate injuries are a type of tear of the metatarsophalangeal joint capsule, occurring most frequently at the distal aspect of the plantar plate, either medial or lateral. When they involve the hallux, they are known as “turf toe”. Most of the lesser plantar plate tears occur in the second metatarsophalangeal joint [29]. They may affect the plantar plate itself, the collateral ligaments, or interosseous tendons [33].

Ultrasound findings include thickening, hypoechogenicity, discontinuity, and entheseal irregularity at the base of the proximal phalanx [30]. Ultrasound has the advantage of allowing dynamic evaluation to the plantar plates, especially assessing its integrity or retraction during dorsiflexion stress of the toes. Ideally, the ultrasound scan should be performed with focused high-frequency probes, such as the “hockey stick” probe.

MRI findings include insertion rupture, retraction, laxity, and redundancy, with inflammatory changes related to the age of the tear. Bone marrow edema in the metatarsal head can be seen due to abnormal motion and biomechanical changes of the forefoot. Since tears usually happen to the lateral or medial distal insertion sites, the flexor tendon and the plantar plate can shift laterally or medially in relation to the metatarsal base [34,44]. Figure 13 presents a case of plantar plate injury with corresponding imaging features.

### 6.10. Freiberg’s Infraction

Freiberg’s infraction is a subchondral fracture that usually affects the second metatarsal head and generally occurs in young patients. They have a multifactorial cause that includes mechanical stress, impaction fracture, vascular insult, and following osteonecrosis and collapse [27].

MRI findings appear before they are seen radiographically and include bone marrow edema, with tiny subarticular and serpentine low signal linear abnormality representing the fracture line, associated with adjacent bone marrow edema. Fragmentation of the articular surface and flattening of the metatarsal head articular surface can be seen in advanced cases [45,46]. Refer to Figure 14 for a typical case of plantar plate injury.

### 6.11. Crystal Arthropathies

Crystals can deposit in and around the joints, with the two main causes described below.

Calcium pyrophosphate deposition disease (CPPD) is a common arthropathy caused by the deposition of the crystals in hyaline cartilage, fibrocartilage, synovium, ligaments, and tendons. Prevalence increases with age, being common and asymptomatic in the elderly. The joints most commonly affected include the knee, pubic symphysis, and wrist [47].

Hydroxyapatite crystal deposition (HADD) disease is caused by the deposition of basic calcium phosphate crystals in joints or periarticular soft tissues, predominantly in tendons and ligaments. It typically occurs in middle-aged patients and can present with acute and severe symptoms, or it may be asymptomatic and incidentally detected on radiographs [47].

MRI is an excellent modality to assess the extent and severity of crystal arthropathies, but the findings may be nonspecific. MRI can be used to assess erosions, chondral defects, and periarticular edema. Calcium hydroxyapatite deposition, for example, appears as hypointense signal areas on both T1 and T2-weighted images. In the acute phase, there may be surrounding soft tissue and joint inflammation, with hyperintense signal in bone and adjacent muscles. Intraosseous migration or rapidly destructive arthritis of joints may occur, characterized by joint destruction, effusions, calcific deposits, and debris [47]. Figure 15 shows a case of crystal arthropathy.

### 6.12. Baxter Neuropathy

Baxter neuropathy is a term that defines nerve entrapment syndrome resulting from the compression of the inferior calcaneal nerve (Baxter nerve), which is the first branch of the lateral plantar nerve. Since it innervates the abductor digiti minimi, imaging findings include edema-like signal intensity changes within the affected muscle and in the long term lead to fatty atrophy [48]. Figure 16 shows a case of Baxter neuropathy.

## 7. Complications

The most feared complication of peripheral venous thrombosis is pulmonary embolism (PE) [1,49]. However, it remains controversial whether PVT can be the initial route of a thromboembolism that extends into the legs and further to the lung [2]. Some authors hypothesize increased risk of PE from plantar thrombosis in non-anticoagulated patients due to repeated compression of the foot muscles [15]. Therefore, patients with simultaneous pulmonary symptoms and foot pain should receive a careful evaluation and undergo US Doppler exams [3]. Even though in the series of 22 cases of PVT by Czihal et al. there were no cases of PE or post-thrombotic syndrome, the authors consider that, as PVT shares common risk factors with DVT, and in accordance with DVT concepts, PVT presents a risk, although very low, of symptomatic PE and post-thrombotic syndrome [7]. It is also important to note that recurrence of venous thromboembolism is not uncommon and is often manifested as PVT [7].

## 8. Treatment

Unlike lower limb superficial and deep venous thrombosis, PVT has no standardized treatment in the literature [1,2]. Some authors believe that treatment with non-steroidal anti-inflammatory drugs alone is sufficient [15], while others advocate the use of anticoagulation therapy for 4 to 6 weeks, combined with knee-length compression stocks [7]. Although scarce, there is some data in the literature suggesting that not using anticoagulants could increase the rate of progression to the leg veins [7]. On the other hand, anticoagulation therapy is associated with increased risk of bleeding in some studies [16]. With regard to thrombophilia screening in DVT, there is a clinical assessment of thrombophilia, recommended for all patients, and thrombophilia testing, recommended for selected cases [50]. Since PVT shares common risk factors with DVT [7], clinical assessment for thrombophilia should also be performed in a similar way to DVT.

In Figure 17, we illustrate a case of PVT with follow-up. A proposed algorithm for diagnosing and managing PVT is presented in Figure 18 based on the literature on the subject and the authors’ experience.

## 9. Conclusions

Plantar vein thrombosis should be included in the differential diagnosis of foot pain, especially when symptoms are unilateral. The plantar veins are usually overlooked in routine US Doppler assessment, thus being an under-diagnosed condition. Diagnosis can easily be made through US and especially MRI due to its typical features, therefore, knowledge of the plantar venous anatomy and active search for such imaging findings are needed.

## Figures and Tables

**Figure 1 diagnostics-14-00126-f001:**
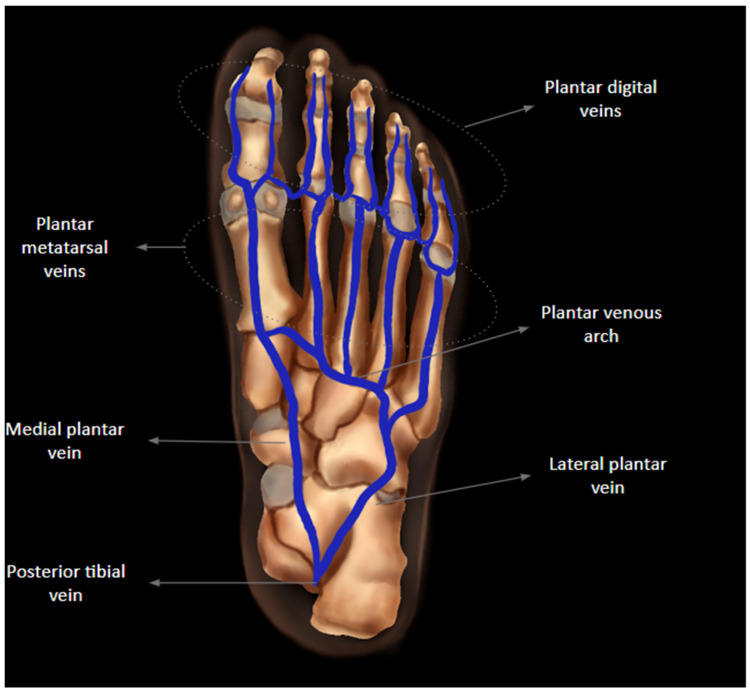
Anatomic illustration of the venous anatomy of the foot, formed by plantar digital veins, plantar metatarsal veins, plantar venous arch, medial plantar vein, lateral plantar vein, and posterior tibial vein.

**Figure 2 diagnostics-14-00126-f002:**
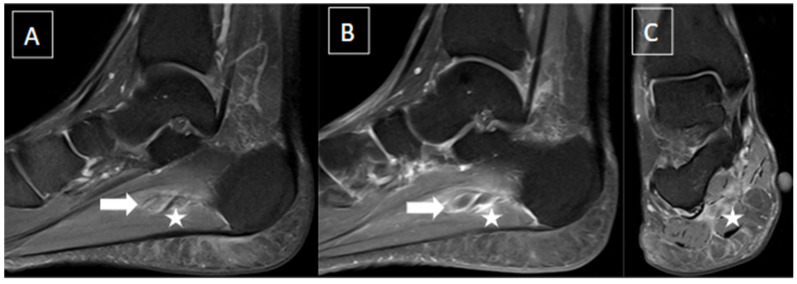
48-year-old female presenting with pain in the plantar aspect of the right ankle for 4 days. Ankle MR images in sagittal T2 fat-suppressed (**A**), post-gadolinium T1 fat-suppressed (**B**), and coronal T2 fat-suppressed (**C**) demonstrate thrombosis of the lateral plantar vein, characterized by venous enlargement with intraluminal thrombus (arrow in (**A**)), venous filling defect (arrow in (**B**)), and perivascular soft tissue edema and enhancement (stars from (**A**) to (**C**)).

**Figure 3 diagnostics-14-00126-f003:**
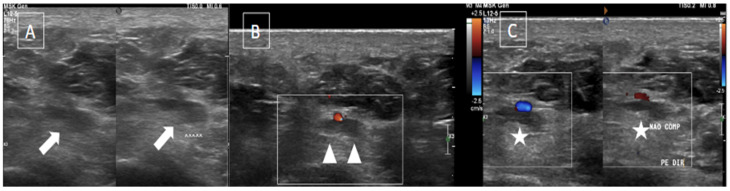
The same patient from Figure 2 underwent an ultrasound with color Doppler showing enlargement of the lateral plantar vein due to an internal thrombus (arrows in (**A**)) and absence of flow on Doppler assessment (arrowheads in (**B**)) associated with loss of compressibility (stars in (**C**)).

**Figure 4 diagnostics-14-00126-f004:**
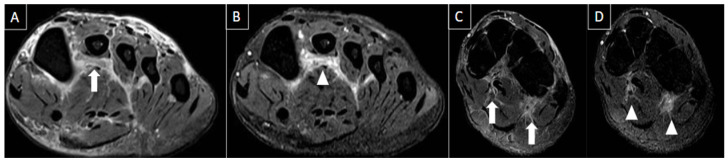
50-year-old female patient with a history of pain in the foot for 12 days. MR images of the forefoot in the short axis (T2 weighted fat-suppressed short axis in (**A**,**C**) and T1 fat-suppressed post-gadolinium short axis in (**B**,**D**)) show acute thrombosis of the plantar venous arch and medial and lateral plantar veins, with perivascular edema (arrows in (**A**,**C**)) and perivascular enhancement and venous filling defects (arrowhead in (**B**,**D**)).

**Figure 5 diagnostics-14-00126-f005:**
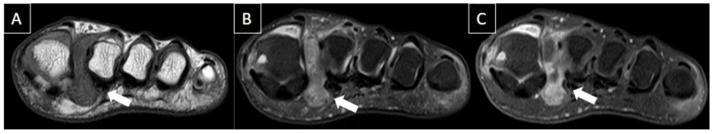
44-year-old female patient with history of rheumatoid arthritis and chronic pain in the left foot. T1-weighted (**A**) and T2-weighted fat-suppressed (**B**) and T1-weighted post-gadolinium (**C**) MR images of the forefoot in the short axis show intermetatarsal bursitis of the 1st interdigital space (arrows), abutting to the plantar fat pad.

**Figure 6 diagnostics-14-00126-f006:**
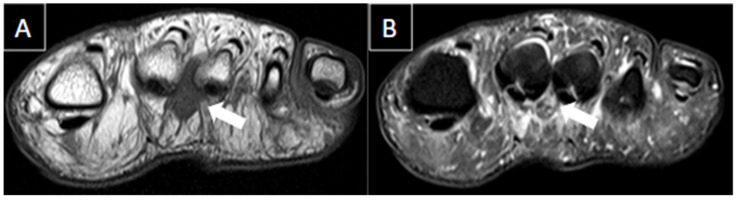
56-year-old female patient with a 6-month history of pain. Short axis T1-weighted (**A**) and T2 fat-suppressed (**B**) MR images show Morton’s neuroma in the plantar aspect of the 2nd intermetatarsal space (arrows).

**Figure 7 diagnostics-14-00126-f007:**
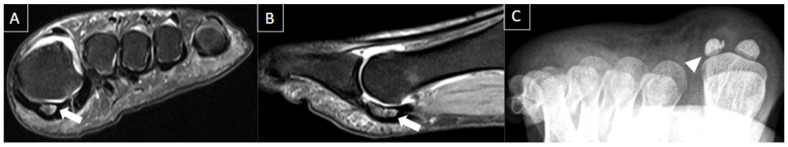
30-year-old female patient with pain in left foot for one month. Short axis T2-weighted fat-suppressed (**A**) and sagittal T2-weighted fat-suppressed (**B**) MR images and axial sesamoid radiographic views (**C**) present bone edema of the medial sesamoid on MRI (arrows) and irregular and sclerotic appearance of the lateral sesamoid with reduced dimensions (arrowheads). Findings are consistent with sesamoiditis.

**Figure 8 diagnostics-14-00126-f008:**
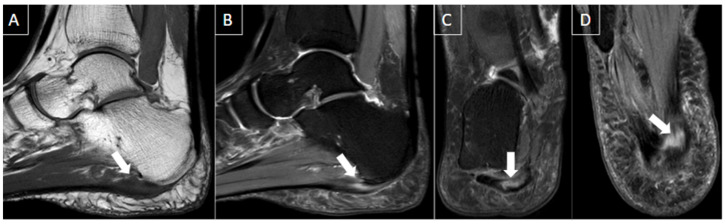
55-year-old female patient with heel pain during walking for 4 months. Ankle MR images in sagittal T1-weighted (**A**), sagittal T2-weighted fat-suppressed (**B**), coronal T2-weighted fat-suppressed (**C**), and axial T2-weighted fat-suppressed (**D**) show significant plantar fasciitis of the proximal central bundle (arrows), characterized by thickening and partial tear of interstitial fibers. Bone edema at the calcaneal attachment and surrounding soft tissue inflammatory changes are also seen.

**Figure 9 diagnostics-14-00126-f009:**
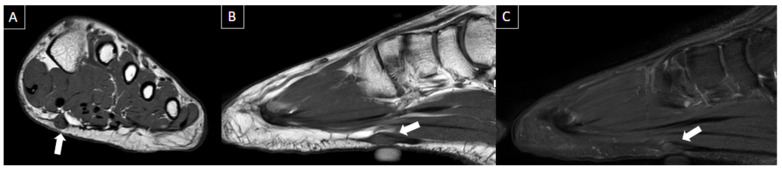
41-year-old male with swelling and pain in the left foot. Short axis T1-weighted (**A**), sagittal T1-weighted (**B**), and sagittal fat-suppressed T2-weighted (**C**) MR images present a fusiform nodule in the central band of the plantar fascia, at the midfoot level (arrows), consistent with plantar fibroma. The nodule correlates with the patient’s symptom location (cutaneous marker).

**Figure 10 diagnostics-14-00126-f010:**
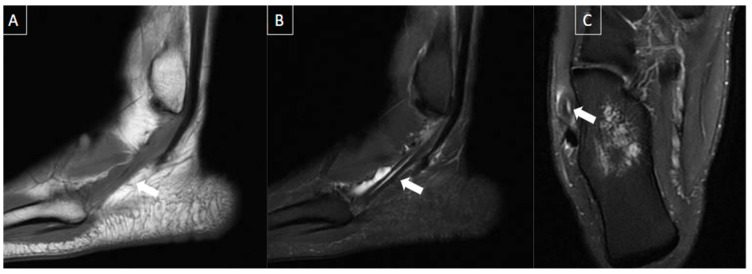
41-year-old male patient with pain in the left foot for four days after intense running exercise. MR images of the forefoot in sagittal T1 (**A**), sagittal T2-weighted fat-suppressed (**B**), and axial T2-weighted fat-suppressed (**C**) show tendinopathy of the retro and inframalleolar segment of peroneous brevis tendon with split tear extending to its insertion at the fifth metatarsal base (arrows).

**Figure 11 diagnostics-14-00126-f011:**
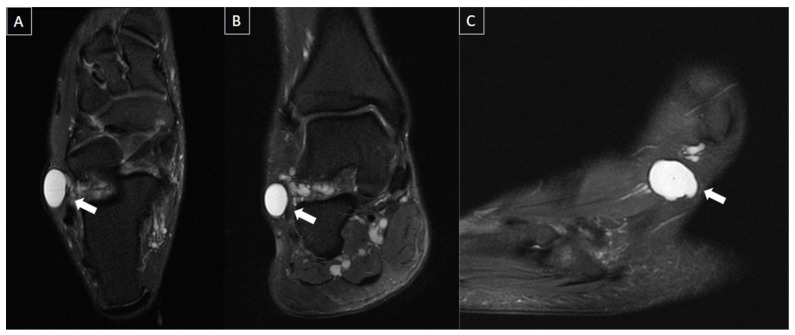
54-year-old male with localized pain in the lateral aspect of the ankle. MR images in axial (**A**), coronal (**B**), and sagittal (**C**) T2-weighted fat-suppressed show multi lobulated and septated cyst in communication with sinus tarsi ligaments and posterior subtalar joint (arrows), extending into the lateral subcutaneous layer, consistent with synovial/ganglion cyst.

**Figure 12 diagnostics-14-00126-f012:**
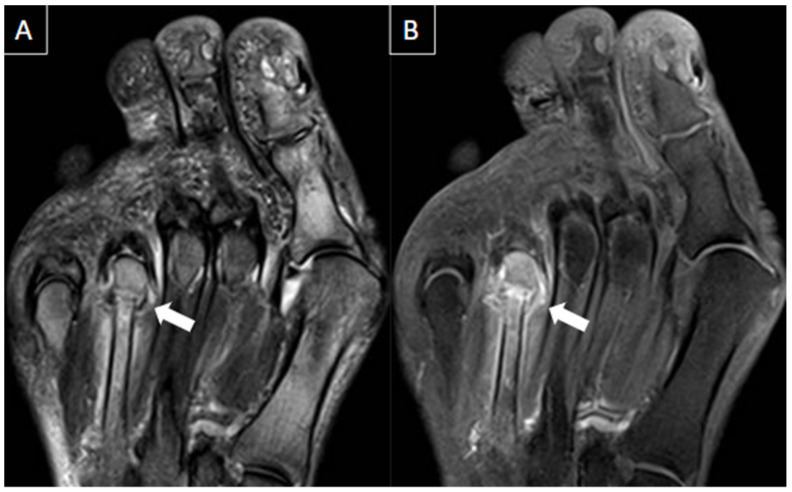
54-year-old female with history of bulimia and recent pain in the left foot after walking. MR images in the long axis (T2-weighted fat-suppressed in (**A**), and T1 fat-suppressed post gadolinium in (**B**)) show stress fracture of the 4th metatarsal neck associated with bone edema, periostitis, and edema of the surrounding soft tissues (arrows).

**Figure 13 diagnostics-14-00126-f013:**
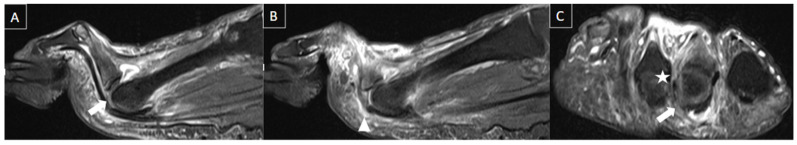
51-year-old male patient with acute forefoot pain during physical activities that happened months ago. MR images in sagittal T2-weighted fat-suppressed (**A**,**B**) and short axis post-gadolinium T1-weighted fat-suppressed (**C**) demonstrate complete tear of the distal insertion of the second metatarsophalangeal plantar plate (arrows), with proximal retraction (arrowhead) with associated lateral collateral ligament (star), with edema and enhancement of adjacent soft tissues.

**Figure 14 diagnostics-14-00126-f014:**
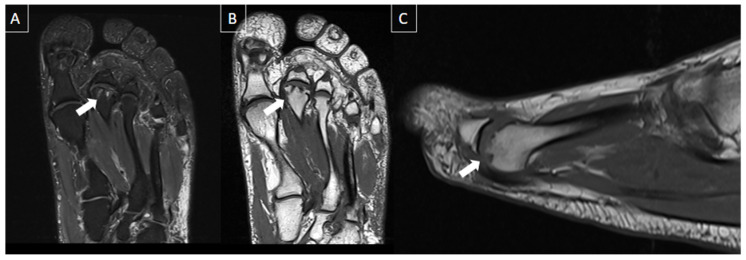
56-year-old male patient with metatarsalgia for 6 months. MR images of the forefoot in the long axis (T2-weighted fat-suppressed (**A**) and T1-weighted (**B**)) and sagittal T1-weighted (**C**) present chronic deformity and depression of the second metatarsal head, with small subchondral cysts (arrows) suggestive of Freiberg’s infraction.

**Figure 15 diagnostics-14-00126-f015:**
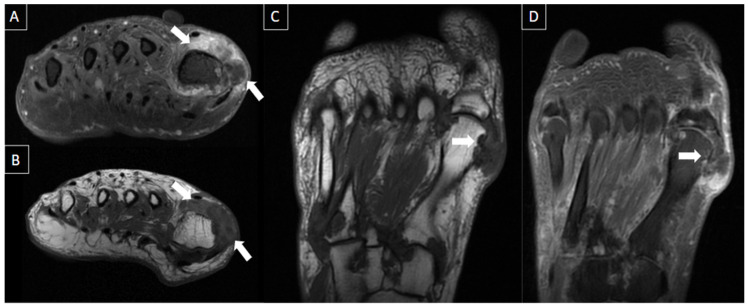
47-year-old male presenting with swelling and pain at the base of greater toe. MR images in short axis T1 fat-suppressed post-gadolinium (**A**), short axis T1-weighted (**B**), long axis T1-weighted (**C**), and T1 fat-suppressed post-gadolinium (**D**) show multiple foci of cortical erosions with edema and contrast enhancement, with associated synovial thickening and amorphous low signal intensity tissue suggestive of crystal deposits (arrows), notably at the metatarsophalangeal joint but also seen at the tarsometatarsal joints. Classic imaging features of gout.

**Figure 16 diagnostics-14-00126-f016:**
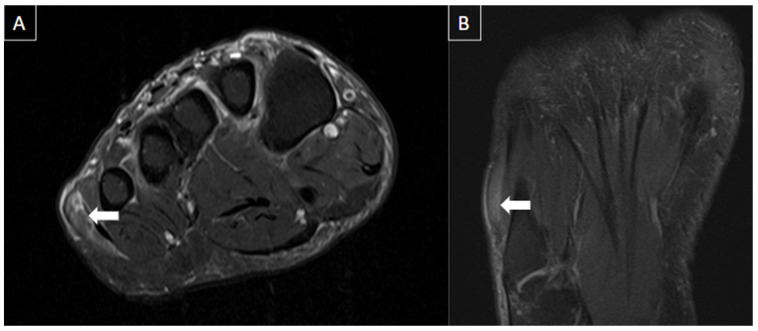
53-year-old female presenting with paresthesia in the lateral plantar region. MR images of the forefoot in T2-weighted fat-suppressed short axis (**A**) and long axis (**B**) show mild edema of the abductor digiti minimi muscle (arrows), suggesting acute denervation due to compressive neuropathy (Baxter’s neuropathy).

**Figure 17 diagnostics-14-00126-f017:**
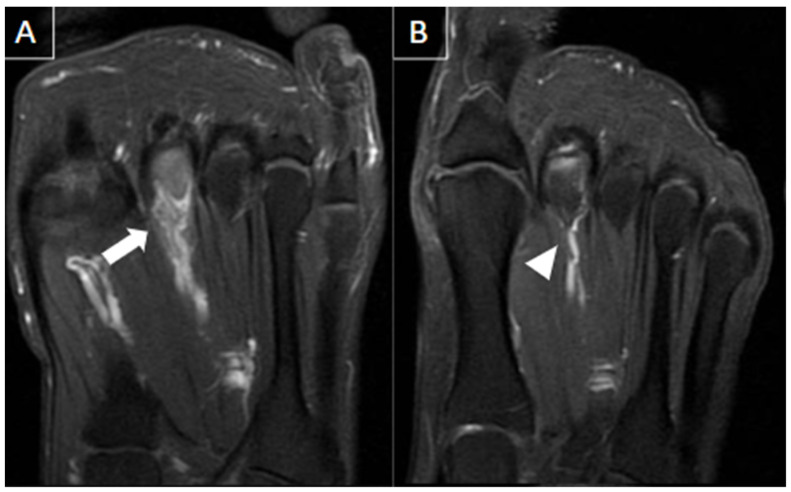
40-year-old female patient with a history of twisting injury of the forefoot 10 days ago. MR images in T1 fat-suppressed post-gadolinium long axis show acute thrombosis of the second metatarsal vein with perivascular edema (arrow in (**A**)); (**B**) signs of recanalization of the thrombosis 1 month later, but still with mild surrounding edema (arrowheads).

**Figure 18 diagnostics-14-00126-f018:**
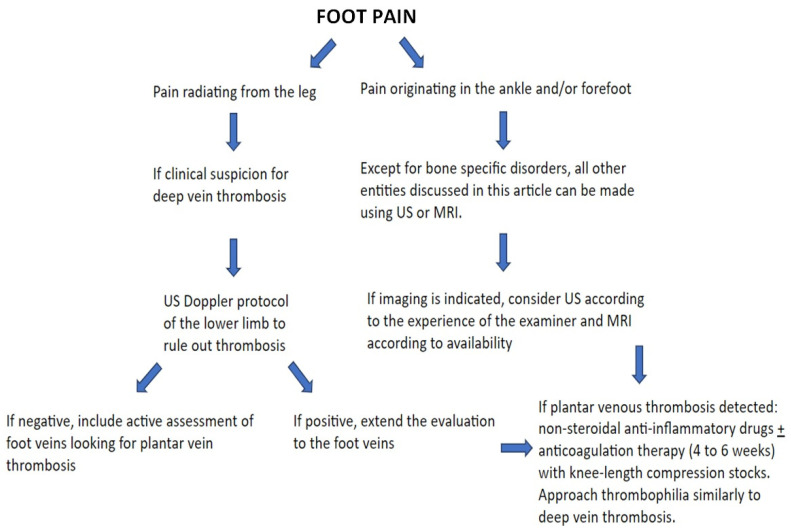
Proposed algorithm on how to approach plantar vein thrombosis using imaging methods.

**Table 1 diagnostics-14-00126-t001:** Imaging findings of the main imaging methods to evaluate plantar venous thrombosis. MRI—magnetic resonance imaging; US—ultrasound.

MRI	B-Mode US	Color and Pulsed-Wave Doppler US
Perivascular edema and enhancement Muscle edema Intraluminal signal change Venous enlargement Presence of collateral veins Venous filling defects (post-gadolinium injection)	Local tenderness Loss of compressibility Venous enlargement Intraluminal content Perivascular edema	Local tenderness Loss of flow Filling defects

**Table 2 diagnostics-14-00126-t002:** Imaging and clinical findings of the differential diagnosis of plantar vein thrombosis. MRI—magnetic resonance imaging; US—ultrasound.

Differential Diagnosis	Clinical	US	MRI
Intermetatarsal bursitis	Metatarsalgia, frequently associated with Morton’s neuroma and plantar plate tears.	Thin-walled bursa distended with hypoechoic fluid and peri bursal hyperemia (acute). Thickened bursal wall, chronic synovial proliferation, more echogenic content, and intrabursal hyperemia (chronic).	Hypointense on T1-weighted and hyperintense on T2-weighted fat-suppressed images. Peripheral enhancement may be seen post gadolinium.
Morton’s neuroma	Forefoot pain which radiates from the midfoot to toes. Symptoms are often progressive and worsened by activity.	Well-defined ovoid mass with variable echogenicity with continuity with the interdigital nerve in the long axis. The mass can be tender and mobile when compressed, with vascularity on power Doppler.	Hypointense to isointense on T1-weighted images and hypointense to hyperintense on T2-weighted fat-suppressed images.
Sesamoiditis	Painful inflammatory/mechanical condition caused by repetitive injury to the plantar aspect of the forefoot.	Not generally used but can show shrinking and fragmentation in chronic cases.	Increased signal intensity on fluid-sensitive sequences due to marrow edema (acute phase). In chronic stages, it can manifest with sclerosis.
Plantar fasciitis	Most common cause of unilateral heel pain.	Ultrasound findings include thickening greater than 4.5 mm (the most useful sign), hypoechogenicity of the plantar fascia, and loss of normal fibrillar reflective echotexture.	Hypointense or isointense fascial thickening at calcaneal insertion on T1-weighted images.
Plantar fibromatosis	Pain due to mass effect or infiltration of adjacent muscles or neurovascular structures.	Fusiform-shaped nodule at the plantar fascia away from the calcaneal insertion, either hypoechoic or isoechoic to the fascia.	Isointense plantar fascia nodule on T1-weighted and T2-weighted fat-suppressed images with contrast enhancement.
Tendon pathologies	Varies according to the cause (tendinosis/ tendinopathy, tenosynovitis, and peritendinitis).	Intra and peritendinous alterations with enlargement of the tendon and various degrees of hypoechogenicity.	Enlargement and signal abnormalities in the affected tendon. High signal intensity can be seen involving the tendon and peritendinous soft tissues.
Ganglion/synovial cysts	Mass effect on adjacent structures (frequently related to trauma history).	Well-defined uni or multilocular fluid-filled anechoic masses with posterior acoustic enhancement.	Mass with water equivalent signal (uniformly hyperintense on T2), and the walls can show post-gadolinium enhancement.
Stress fractures	Most common bony cause of metatarsalgia.	Thickening and hypervascularity of the periosteum, cortical irregularities, and soft tissue edema.	Periosteal, bone marrow, and soft tissue edema.
Plantar plate injuries	Metatarsalgia and/or deformity in cases of full-thickness plantar plate tears.	Thickening and hypoechogenicity or discontinuity and entheseal irregularity at the base of the proximal phalanx.	Ligament rupture, thickening, laxity, and redundancy. Bone marrow edema of the metatarsal head and phalanx base due to mechanical changes.
Freiberg’s infraction	Pain on weight-bearing with swelling and tenderness.	Not generally used but can show deformity and depression of the metatarsal head in advanced cases.	Linear subchondral hypointense line on T1 and T2-weighted images with surrounding bone marrow edema.
Crystal arthropathies	Severe acute or subacute pain, swelling, erythema, and warmth of one or more joints and is usually self-limited.	Periarticular edema.	Erosions, chondral defect, and periarticular edema. Hypointense signal intensity areas on both T1 and T2-weighted images (hydroxyapatite deposition).
Baxter neuropathy	Pain, related to the entrapment of the inferior calcaneal nerve.	Edema within the affected muscle.	Edema, atrophy, and fatty infiltration, according to the denervation stage

## Data Availability

Not applicable.

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
