# Peer review of "Imaging Features of Plantar Vein Thrombosis: An Easily Overlooked Condition in the Differential Diagnosis of Foot Pain"

_diagnostics, 2024, doi:10.3390/diagnostics14020126_

Round 1
Reviewer 1 Report
Comments and Suggestions for Authors
A review of the imaging findings of plantar vein thrombosis and the differential diagnosis with images of pathologies that could be confused with plantar vein thrombosis in forefoot is presented.
The review is well written and clear in the presentation of the content.
In my opinion, it should included a section on literature search methodology with a PRISMA flow diagram and a table of the papers included and their characteristics.
In the section 6. Differential diagnosis, there are several comments:
The aim of the review is the differential diagnosis in metatarsalgia the figures must be from the forefoot. (Figures 8, 10 and 11)
The inclusion of plantar fasciitis and Baxter’s neuropathy in this review is unclear.
Author Response
Reviewer 1
Q: A review of the imaging findings of plantar vein thrombosis and the differential diagnosis with images of pathologies that could be confused with plantar vein thrombosis in forefoot is presented.
A: Thank you for the kind words.
Q: The review is well written and clear in the presentation of the content.
A: Thank you for the kind words.
Q: In my opinion, it should included a section on literature search methodology with a PRISMA flow diagram and a table of the papers included and their characteristics.
A: Thank you for the valuable comment. However, differently from a systematic review, we conducted a narrative (descriptive) review of the literature. For this reason, we didn't follow the PRISMA methodology. PRISMA (Preferred Reporting Items for Systematic Reviews and Meta-Analyses) is an evidence-based minimum set of items aimed for systematic reviews and meta-analyses. Since the subject of plantar vein thrombosis is scarce in the literature, we included all the relevant articles on the subject, as well as their main differential diagnoses and imaging aspects.
Q: In the section 6. Differential diagnosis, there are several comments: The aim of the review is the differential diagnosis in metatarsalgia the figures must be from the forefoot. (Figures 8, 10 and 11)
A: Thank you for this valuable comment. Regarding the differential diagnoses of plantar vein thrombosis, there are entities that affect the hindfoot that are recognized in the literature as important differentials, such as plantar fasciitis. Thus, these differential diagnoses also extend beyond metatarsalgia. We agree that because the title includes the word metatarsalgia, the presence of hindfoot differentials can leave the reader confused. Therefore, we removed the word metatarsalgia from the title and kept the important hindfoot differentials throughout the text and the new title became: "Imaging features of plantar vein thrombosis: an easily overlooked condition in the differential diagnosis of foot pain".
Q: The inclusion of plantar fasciitis and Baxter’s neuropathy in this review is unclear.
A: Thank you for the valuable comment. As explained in the question above, we have changed the title of the article so that the reader understands
differentials are not limited to the forefoot, but to the entire foot.
Reviewer 2 Report
Comments and Suggestions for Authors
Dear Authors!
I appreciate your efforts in summarizing data on this subject. I read the paper with a great interest and I believe it would be of real help for those involved in a VTE management.
I have just some minor remarks
Lines 92-93. Virchow's triad is not a pathology.
Line 121. This statement is more or less true only for isolated distal DVT but not for deep vein thrombosis in general. This is what was written in a review that you cited. It's better to reference not to this review but to the study about D-dimer in distal DVT
Line 133. Better say venous, not vascular compressibility
Line 164, 528, 529. Better say pulmonary embolism
Line 200 and Table 1. For venous specialists Doppler term is usually referred to investigation with only graphical information and sound. Could you use another term?
Comments on the Quality of English Language-
Author Response
Reviewer 2
Q: I appreciate your efforts in summarizing data on this subject. I read the paper with a great interest and I believe it would be of real help for those involved in a VTE management.
A: Thank you for the kind words.
Q: I have just some minor remarks. Lines 92-93. Virchow's triad is not a pathology.
A: We have changed the word “pathology” for “factor”.
Q: Line 121. This statement is more or less true only for isolated distal DVT but not for deep vein thrombosis in general. This is what was written in a review that you cited. It's better to reference not to this review but to the study about D-dimer in distal DVT.
A: Thank you for the valuable comment. Since this is more or less true only for isolated distal DVT but not for deep vein thrombosis in general, we have chosen to remove this statement from the article, to avoid any confusion to the reader.
Q: Line 133. Better say venous, not vascular compressibility
A: Thank you for the suggestion. We've made the substitutions for "venous".
Q: Line 164, 528, 529. Better say pulmonary embolism.
A: Thank you for the suggestion. We've made the substitutions for " pulmonary embolism".
Q: Line 200 and Table 1. For venous specialists Doppler term is usually referred to investigation with only graphical information and sound. Could you use another term?
A: Thank you for the valuable comment. In line 200, we changed the term Doppler to duplex ultrasound, as the concept of the sentence involves B-mode and Doppler evaluation. Regarding the table, we made it clearer by adding the term B-mode to US in the middle column and the terms color and pulsed-wave to refer to Doppler in the third column.